# ITEM3D: Illumination-Aware Directional Texture Editing for 3D Models

## Abstract

Texture editing is a crucial task in 3D modeling that allows users to automatically manipulate the surface properties of 3D models. However, the inherent complexity of 3D models and the ambiguous text description lead to the challenge in this task. To address this challenge, we propose ITEM3D, an illumination-aware model for automatic 3D object editing according to the text prompts. Leveraging the power of the diffusion model, ITEM3D takes the rendered images as the bridge of text and 3D representation, and further optimizes the disentangled texture and environment map. Previous methods adopt the absolute editing direction namely score distillation sampling (SDS) as the optimization objective, which unfortunately results in the noisy appearance and text inconsistency. To solve the problem caused by the ambiguous text, we introduce a relative editing direction, an optimization objective defined by the noise difference between the source and target texts, to release the semantic ambiguity between the texts and images. Additionally, we gradually adjust the direction during optimization to further address the unexpected deviation in the texture domain. Qualitative and quantitative experiments show that our ITEM3D outperforms SDS-based methods on various 3D objects. We also perform text-guided relighting to show explicit control over lighting.

## 1 Introduction

Texture editing is an important task in 3D modeling that involves manipulating the surface properties of 3D models to create a visually fantastic and appealing appearance according to the user's ideas. With the increasing applications of 3D models in entertainment and e-shopping, how to automatically generate and edit the texture of a 3D model without manual effort becomes an appealing task in the field of 3D vision. However, this task is challenging due to the complexity of 3D models and the special representation of the texture.

To sufficiently handle the above applications, it would be desirable if a texture editing method can fulfill the following aspects: 1) **Realism**: The generated textures should give rise to realistic and visually natural 2D images after rendering. It requires generative models to capture the complex patterns and structures present in the textures of the 3D model. 2) **Relighting**: The relighting ability allows adjusting the lighting conditions of the edited model to be consistent with the changes made to its texture. 3) **Efficiency**: Texture editing should be efficient and scalable. This requires the use of fast and memory-efficient generative models that can generate high-quality textures in a short time.

Recent advances have demonstrated the effectiveness of generative models in synthesizing high-quality textures that are both visually pleasing and semantically meaningful. The use of generative adversarial networks (GANs) (50; 2; 43; 7) has shown promising results in producing textures with intricate patterns and complex structures. Other approaches, such as texture synthesis via direct optimization (8; 9; 40; 54) or neural style transfer (3; 14; 48; 24), have also been explored for their

ability to generate textures with specific artistic styles. However, the capacity of these models is still unable to meet the need of real-world applications, which requires high-quality and diverse textures. Meanwhile, recent researches (6; 34; 25; 20; 21; 17; 23) on the diffusion models have emerged as a powerful new family of generative methods, which achieve impressive generation results in natural images and videos, inspiring us to introduce the awesome power into the task of 3D modeling.

However, directly applying the diffusion model to 3D objects is a non-trivial task due to the following reasons. 1) **The gap between the 3D representation and natural images.** Existing diffusion models are typically trained with natural images, making the pre-trained diffusion model lack prior knowledge in the 3D domain. Moreover, due to the complexity of the 3D model, it would be difficult to simultaneously edit shape, appearance, and shading, sometimes leading to conflicts in optimization goals. Therefore, directly editing the 3D representation may cause extreme semantic bias and destruction of inherent 3D topology. 2) **The learning misdirection of text description.** It is hard for text prompts to exactly describe the target images at the pixel level, leading to an ambiguous direction when taking the rendered images as the bridge.

To solve these problems, we present an efficient model, dubbed ITEM3D, which can generate visually natural texture corresponding to the text prompt generated by users. Instead of directly applying the diffusion model for texture editing in the 2D space, we adopt rendered images as the intermediary that bridges the text prompts and the appearance of 3D models. Apart from the appearance, the lighting and shading are also key components influencing the rendering results. Therefore, we represent the 3D model into a triangular mesh and a set of disentangled materials consisting of the texture and an environment map using nvdiffrec (29), which achieves a balance for representing both appearance and shading.

To optimize a texture and an environment map with the diffusion model, a naive idea is to adopt the score distillation sampling (SDS) like 2D diffusion-based editing methods, which represents the absolute direction. Unfortunately, the absolute direction often leads to noisy details and an inconsistent appearance, due to the ambiguous description of the text prompt for the target images. Inspired by the recent improvement (13), we replace the absolute editing direction led by the score distillation sampling with a relative editing direction determined by two predicted noises under the condition of the source text and the target text respectively, as illustrated in Fig. 1 (a). In this way, our model enables us to edit the texture in obedience to the text while bypassing the inconsistency problem by releasing the ambiguous description. It is ideal that the intermediate states between the source and target text can give relatively accurate descriptions for arbitrary rendered images during the optimization, like the green straight lines in Fig. 1 (b). However, the optimization in the texture domain actually shows an unexpected offset of the appearance in rendered images, leading to the deviation from the determined direction, like the red line in Fig. 1 (b). To reduce the deviation caused by the texture projection, we gradually adjust the editing direction during the optimization, as green fold lines shown in Fig. 1 (b). With the advent of the textural-inversion model, it can be easy to automatically correct the description as the change of the texture and its rendered images.

Thanks to the proposed solutions, our method overcomes the challenges of domain gap and learning misdirection, fulfilling all three requirements of texture editing. In summary, our contributions are:

- We design an efficient optimization pipeline to edit the texture and environment map obedient to the text prompt, directly empowering the downstream application in the industrial pipeline.
- We introduce the relative direction to the 3D texture optimization, releasing the problem of noisy details and inconsistent appearance caused by the semantic ambiguity between the texts and images.
- We propose to gradually adjust the relative direction guided by the source and target texts which addresses the unexpected deviation from the determined direction caused by the texture.

## 2 Related Work

**3D Model Representation.** From the perspective of 3D representations, traditional methods typically exploit point clouds or meshes to estimate depth maps (1; 38; 11) or employ a voxel grid and estimate the corresponding occupancy and color (39; 4). However, these methods are often limited to the memory requirement, which results in excessive runtime. With the development of computer vision, neural implicit representations are brought up and leverage differentiable rendering to reconstruct

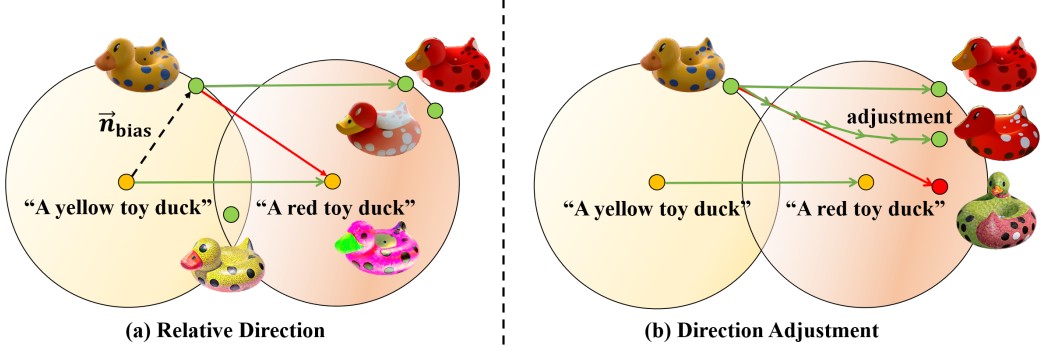

**(a) Relative Direction**                     **(b) Direction Adjustment**

Figure 1: **Motivation**. (a) Previous methods (31; 6) with SDS Loss to directly guide the optimization leads to ambiguous details due to the bias between texts and images (red line), while our method introduces the relative direction between source and target texts to the optimization process, eliminating the bias and improving the rendering results (green line). (b) The optimization in the texture domain gives rise to the deviation of the target direction (red line), thus we gradually adjust the direction to fine-tune the optimization (fold green line).

3D geometry with appearance. Neural Radiance Field (27) and followup methods (52; 47; 28; 5; 49; 51; 33; 22), utilize volumetric representations and a neural encoded field to compute radiance by ray marching. While these NeRF-based methods synthesize high-fidelity rendering results, the quality of the generated geometry is limited due to the ambiguity of volume rendering. Meanwhile, surface-based methods (30; 46; 10) optimize the underlying surface directly. These methods usually rely on volumetric representation and utilize an implicit surface by converging the volumetric representation (30) or constructing a field that converse SDF into density (46; 10). Though surface-based methods achieve better geometry than NeRF-based methods, they require excessive computation runtime since they rely too much on the ray-marching mechanism. Apart from implicit neural representations, there also exist approaches that utilize explicit surface representations to estimate explicit mesh from images. To extend such methods that originally built upon a fixed mesh topology, DMTet (41) employs a differentiable marching tetrahedral layer and optimizes the surface mesh directly. Nvdiffrec (29) further extends DMTet by jointly optimizing mesh topology, materials, and lighting. ITEM3D leverages an explicit mesh representation and optimizes texture and environment map. By supporting the decomposition of shape, materials and lighting, ITEM3D supports texture editing while preserving the topology by design. Additionally, ITEM3D employs an efficient differentiable rasterization pipeline for faster optimization.

**2D Diffusion-based Image Editing.** Owing to the remarkable generalization ability of the diffusion model, a growing number of works (23; 17; 35; 36; 42) emerged to create customized images with specific styles or objects, as well as stunning images based on text descriptions. All these methods rely on the diffusion process by either fine-tuning the diffusion model or refining the target embedding to reach the desired image domain. SDEdit (23) denoises the noisy image through a diffusion process under the given description. DDIB (42) first converts the input image into a latent representation using origin text and subsequently translates the latent into the desired image with the target text. ControlNet (53) trains a controlling module to augment images with additional conditions that improve the controllability of the editing process. DiffusionCLIP (18) fine-tunes the diffusion model, which translate the image from a pretrained domain to a target text domain. Imagic (17) fine-tunes both the text embeddings and the diffusion model to ensure more stable editing. Unlike these methods that optimize in 2D image space, our ITEM3D utilizes the pre-trained diffusion model as a prior for 3D texture optimization.

**3D Text-guided Generation.** With the advent of large text-image models, *i.e.*, the CLIP, recent works (45; 37; 16; 15; 26) have made impressive progress on 3D text-driven synthesis. The majority of methods adopt the optimization procedures supervised by the CLIP similarity (32). Specifically, CLIP-NeRF (45) proposes a unified framework to manipulate NeRF, guided by a text prompt or an example image. Similarly, CLIP-Mesh applies the explicit textured mesh as a 3D representation, able to deform the shape along with its texture corresponding to the text. Apart from the CLIP-based method, the diffusion model (35) recently inspires huge breakthroughs in 3D text-guided generation.

127 Latent-NeRF (25) utilizes the score distillation sampling to bring the NeRF representation to the
128 latent space, showing impressive generation results of the combination between diffusion model and
129 NeRF. TEXTure (34) takes an iterative scheme to paint a 3D model from different viewpoints based
130 on a pre-trained depth-to-image diffusion model. Fantasia3D (6) leverages the disentangled modeling
131 and learns the geometry and appearance supervised by the score distillation sampling. However,
132 these SDS-based methods often produce non-detailed and blurry outputs due to noisy gradients. In
133 contrast, our ITEM3D uses the relative direction to eliminate the semantic ambiguity of the target
134 prompt towards the texture.

## 3 Method

### 3.1 Overview

137 Given a set of multi-view images $\mathcal{I} = \{I_1, ..., I_n\}$, we aim to reconstruct the 3D model with both
138 geometry and texture, and then edit the texture under the guidance of text prompts. To this end, we
139 design a zero-shot differentiable framework that optimizes the disentangled materials of the object,
140 *i.e.*, texture map and environment map. We first leverage a differentiable rendering model $\mathcal{R}$ to
141 represent the 3D model as an accurate shape and surface materials with texture and environment map
142 Sec. 3.2). For further editing of appearance, we utilize the diffusion model to guide the direction
143 of the texture optimization given the target text prompt. To solve the problem of ambiguous and
144 noisy details, we introduce the relative direction of source text and target text into the optimization
145 (Sec. 3.3). Moreover, we gradually adjust relative direction to address the challenges of deviation
146 caused by the unbalanced optimization in the texture domain (Sec. 3.4). The overview of our method
147 is demonstrated in Fig. 2.

### 3.2 3D Model Representation

149 To accomplish editing the appearance of the 3D model via text prompt, we disentangle the 3D model
150 into a triangular mesh and a set of spatially varying materials. The disentanglement thus allows us
151 to edit the texture directly while keeping the geometry invariant. The material model we employed
152 combines a diffuse term, a specular term and a normal term. A four-channel texture is provided for the
153 diffuse parameters $k_d$, where the optional fourth channel $\alpha$ represents the transparency. Meanwhile,
154 the specular term is described by a roughness factor $r$, a metalness factor $m$ and a sheen factor $o$
155 that is unused in our model. These values $(o, r, m)$ are stored in another texture map $k_{orm}$. The
156 normal term in our representation is a tangent space normal map $n$, which is utilized to capture the
157 high-frequency details of the appearance. In order to handle texturing effectively during optimization,
158 we utilize volumetric texturing and access our texture by the world space position $x$. We tackle the
159 challenge of the impractical cubic growth in memory usage of volumetric textures for our target
160 resolution by leveraging a multi-layer perceptron (MLP) to encode the material parameters into a
161 compact representation. Specifically, given a world space position $x$, we compute the base color,
162 $k_d$, the specular parameters, $k_{orm}$ and a tangent space normal perturbation $n$, the mapping is thus
163 formulated as $x \rightarrow (k_d, k_{orm}, n)$. With the introduction of this mapping, for a fixed topology, the
164 textures are initialized by sampling the MLP on the mesh surface and then optimized efficiently.
165 Following the rendering equation of the image-based lighting model, we compute the radiance $L$ in
166 direction $\omega_o$ by:

$$L(\omega_o) = \int_\Omega L_i(\omega_i) f(\omega_i, \omega_o)(\omega_i \cdot \mathbf{n}) d\omega_i, \tag{1}$$

167 where $L_i$ is the incident radiance from direction $\omega_i$, the $f$ is the BSDF and $n$ is the intersection
168 normal of the corresponding integral domain $\Omega$. Specifically, we adopt the Cook-Torrance microfacet
169 specular shading model as the BSDF in our rendering equation:

$$f(\omega_i, \omega_o) = \frac{D \, G \, F}{4(\omega_o \cdot \mathbf{n})(\omega_i \cdot \mathbf{n})}. \tag{2}$$

170 The term $D$ here represents the GGX (44) normal distribution while the term $G$ is the geometric atten-
171 uation and $F$ is the Fresnel term respectively. Furthermore, we employ the split-sum approximation
172 for its efficiency and the rendering equation Eq. (1) can be formulated as:

$$L(\omega_o) \approx \int_\Omega f(\omega_i, \omega_o)(\omega_i \cdot \mathbf{n}) d\omega_i \int_\Omega L_i(\omega_i) D(\omega_i, \omega_o)(\omega_i \cdot \mathbf{n}) d\omega_i. \tag{3}$$

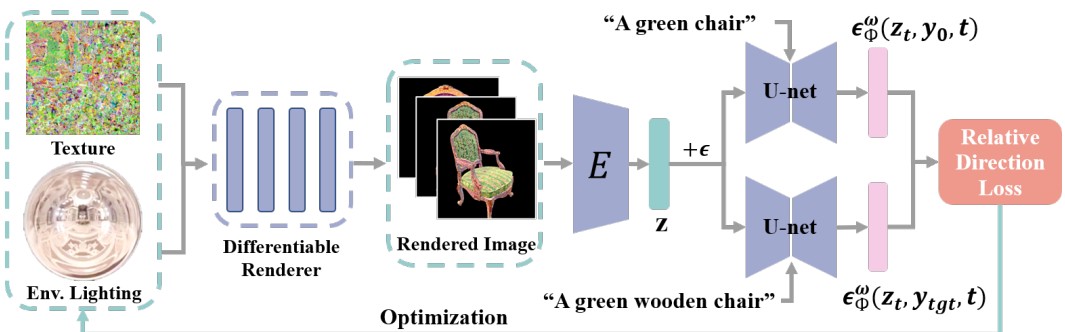

Figure 2: **Pipeline** of texture editing. We render the 3D model with mesh, texture, and environment map into 2D images which are then added with noise $\epsilon$. We then use the source text and the target text as the conditions to denoise via two U-Nets. The difference between the two predicted noises serve as the relative direction to guide the optimization of the materials of the 3D model, *i.e.*, texture and environment map.

The first term of this product only relies on the parameters $(\omega_i \cdot \mathbf{n})$ and the roughness $r$ of the BSDF, which are precomputed and stored in a 2D lookup texture. Meanwhile, the second term is the integral of the radiance with the specular normal distribution function $D$ expressed in Eq. (2), which is also precomputed and stored by a filtered cubemap. Owing to the precomputation and lookup mechanism, the rendering process is then accelerated. In order to learn the environment lighting from 2D image observations, we employ a differentiable shading model to represent this split-sum approximation. The cube map in our case can be represented as trainable parameters, which are initialized as preintegrated lighting.

### 3.3 Relative Direction Based Optimization

Our goal is to enable users to edit the appearance of 3D models using natural language descriptions. To accomplish this, the directional idea is to utilize the diffusion model that has been pre-trained in 2D images as knowledge prior to guide the editing of texture. Naively, we could use Score Distillation Sampling (SDS) loss,

$$\nabla_\theta \mathcal{L}_{\text{SDS}}(\phi, \mathbf{x} = \mathcal{R}(\theta)) = \mathbb{E}_{t,\epsilon}\left[w(t)\left(\epsilon_\phi^\omega(\mathbf{z}_t; y, t) - \epsilon\right)\frac{\partial \mathbf{x}}{\partial \theta}\right], \tag{4}$$

where $\mathbf{x}$ is the rendered images, $t$ is the sampled time step, $z_t$ is the $t$ time step latent, $w(t)$ is the weighting function that equals $\partial \mathbf{z}_t / \partial \mathbf{x}$, $y$ is the text condition, $\epsilon_\phi^\omega(\mathbf{z}_t; y, t)$ is the predicted noise through classifier-free guidance, and $\epsilon \in N(0, I)$ is the noise added to the rendered images. The gradient of SDS loss gives an editing direction for our texture optimization, determined by the text prompt $y$. However, the SDS loss may cause the destruction of original image content with noisy details, because the text prior typically cannot faithfully reflect the information of the image. It is known that the entropy of an RGB image is significantly larger than that of a text prompt. As a consequence, the misdescription inevitably arises when taking the text prompt as the prior to restore the high-quality image from the same-scale noise. Therefore, even for a text prompt $y_0$ describing the original images, there exists a deviation related to the optimized texture $\theta$ between the added noise $\epsilon$ and the predicted noise $\epsilon_\phi^\omega(\mathbf{z}_t; y_0, t)$, which can be simply expressed as,

$$D_{\text{bias}}(\theta, ...) \propto ||\epsilon_\phi^\omega(\mathbf{z}_t; y_0, t) - \epsilon||. \tag{5}$$

Thus, the gradient leads to a bias term from the original input image, which can be expressed as,

$$\vec{n}_{\text{bias}} = \frac{\partial D_{\text{bias}}(\theta, ...)}{\partial \theta} = \left(\epsilon_\phi^\omega(\mathbf{z}_t; y_0, t) - \epsilon\right)\frac{\partial \mathbf{x}}{\partial \theta}. \tag{6}$$

Moreover, for an arbitrary text prompt $y_{\text{tgt}}$ describing the target editing texture, it could be considered that there exists a term of expected editing direction and a term of bias discussed above,

$$\left(\epsilon_\phi^\omega\left(\mathbf{z}_t; y_{\text{tgt}}, t\right) - \epsilon\right) \frac{\partial \mathbf{x}}{\partial \theta} = \vec{n}_{\text{tgt}} + \vec{n}_{\text{bias}}. \tag{7}$$

As a result, the $\vec{n}_{\text{bias}}$ gives rise to the misdirection for the optimization procedure.

To address these issues, it is ideal to find the accurate editing direction $\vec{n}_{\text{tgt}}$, while the term of $\vec{n}_{\text{bias}}$ is hard to estimate due to the diverse input images. To mitigate the gap, it is natural to take the text guidance as a relative direction rather than an absolute direction, enabling us to eliminate the term of $\vec{n}_{\text{bias}}$. The absolute direction of the source $\vec{n}_{\text{src}}$ and the target $\vec{n}_{\text{tgt}}$ can be expressed as,

$$\vec{n}_{\text{src}} = \left(\epsilon_\phi^\omega\left(\mathbf{z}_t; y_0, t\right) - \epsilon\right) \frac{\partial \mathbf{x}}{\partial \theta} - \vec{n}_{\text{bias}}, \tag{8}$$

$$\vec{n}_{\text{tgt}} = \left(\epsilon_\phi^\omega\left(\mathbf{z}_t; y_{\text{tgt}}, t\right) - \epsilon\right) \frac{\partial \mathbf{x}}{\partial \theta} - \vec{n}_{\text{bias}}, \tag{9}$$

where $\vec{n}_{\text{src}}$ is actually the $\vec{0}$ giving no extra information to the input images. Inspired by the CLIP-directional loss improved by the StyleGAN-Nada (12) and the denoising loss proposed by the recent work (31), we utilize the difference between the source $\vec{n}_{\text{src}}$ and the target $\vec{n}_{\text{tgt}}$ as the relative direction of the target, which can be presented as,

$$\vec{n}_{\text{tgt}} = \vec{n}_{\text{tgt}} - \vec{n}_{\text{src}} = \left(\epsilon_\phi^\omega\left(\mathbf{z}_t, y_{\text{tgt}}, t\right) - \epsilon_\phi^\omega\left(\mathbf{x}, y_0, t\right)\right) \frac{\partial \mathbf{x}}{\partial \theta}. \tag{10}$$

Therefore, the final gradient utilized for optimizing the texture can be presented as,

$$\nabla_\theta \mathcal{L}_{\text{RDL}}(\phi, \mathbf{x} = \mathcal{R}(\theta)) = \mathbb{E}_{t,\epsilon}\left[w(t)\left(\epsilon_\phi^\omega(\mathbf{z}_t; y_{\text{tgt}}, t) - \epsilon_\phi^\omega(\mathbf{z}_t; y_0, t)\right) \frac{\partial \mathbf{x}}{\partial \theta}\right], \tag{11}$$

## 3.4  Direction Adjustment

Different from the gradual transition in the nature image domain, the optimization of the texture domain unfortunately shows an unexpected offset of the appearance in rendered images, due to the complex projection in differentiable rendering. The inherent reason is that the complexity of rendering leads to unbalanced optimization for the texture, with some parts under-tuning and other parts over-tuning. This appearance offset can be seen in some parts of the rendered image, leading to the inconsistency between the source text and the rendered images in the median period of the optimization procedure. It is known that a source image with an inconsistent text description means an optimization misdirection which leads to an unknown change in the editing results. Similar to the known problem, if a rendered image during the median optimization hops out the direction between the source text and the target text, it can be considered as the inconsistent description for the source image when we take the current median point as a relative beginning point. The original editing direction is give by,

$$\vec{n}_{\text{ori}} = \vec{n}_{\text{tgt}} - \vec{n}_{\text{src}}. \tag{12}$$

If the optimization continues along the original direction, a more severe deviation can be attached to the optimization procedure.

To avoid the misdirection caused by the texture domain, we propose to adjust the editing direction, specifically the source text prompt, during our optimization process of the texture map. The adjusted direction $\Delta \hat{T}_i$ can be represented as,

$$\Delta \hat{T}_i = \vec{n}_i - \vec{n}_{i-1} = \mathcal{B}(I_i) - \mathcal{B}(I_{i-1}), \tag{13}$$

where $i$ is the optimization iteration and $\mathcal{B}(\cdot)$ expresses the inverse text generated by a pre-trained language-image model BLIP-v2 (19).

As shown in the Fig. 1 (b), the direction is continually adjusted during the optimization so that the new global direction $\vec{n}_i$ can be written as,

$$\vec{n}_i = \vec{n}_{\text{ori}} + \sum_{j \leq i} \Delta \hat{T}_j. \tag{14}$$

By adjusting the optimization direction step by step, we achieve more delicate and controllable editing, which can be seen in Sec. 4.3.

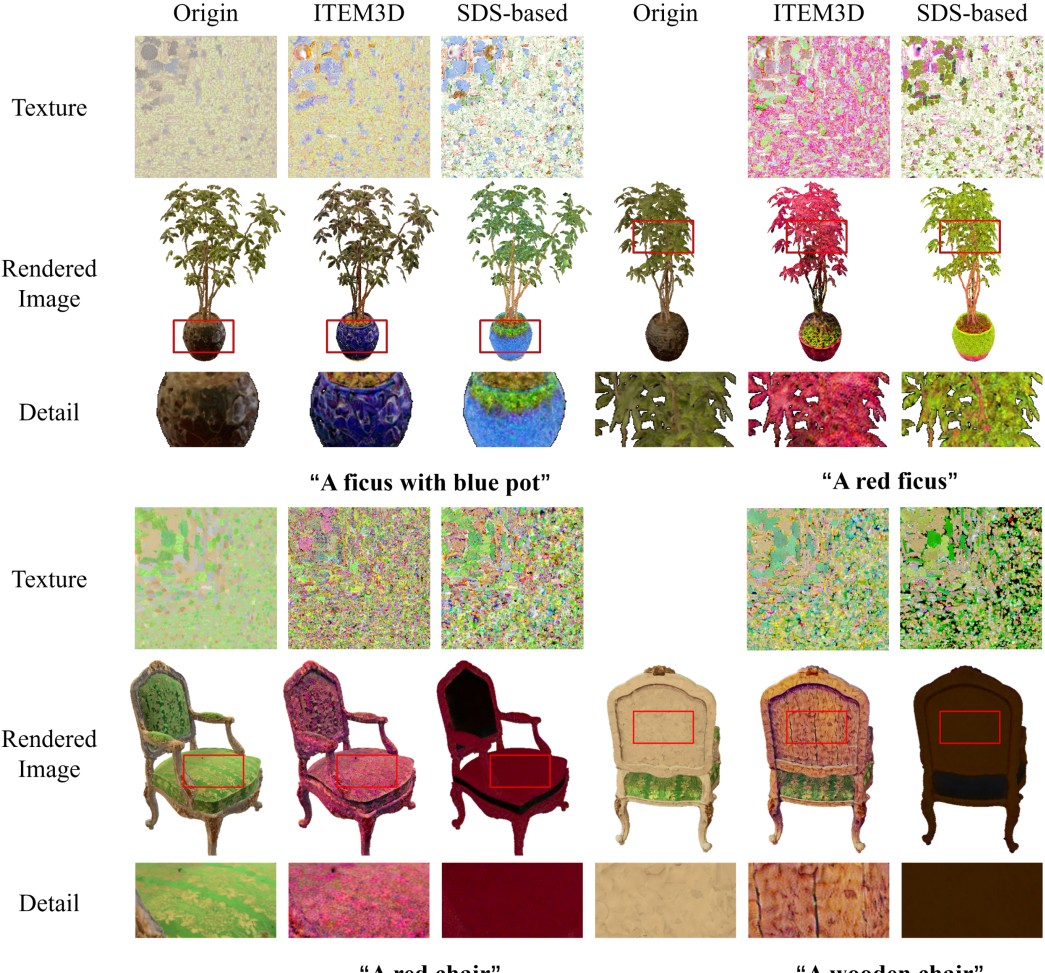

|        | Origin | ITEM3D | SDS-based | Origin | ITEM3D | SDS-based |
|--------|--------|--------|-----------|--------|--------|-----------|

Texture

Rendered Image

Detail

**"A ficus with blue pot"**          **"A red ficus"**

Texture

Rendered Image

Detail

**"A red chair"**          **"A wooden chair"**

Figure 3: **Qualitative comparison** on NeRF synthetic dataset. The results of both textures and rendered images are presented. Our method synthesizes more realistic objects which better correspond to text instructions.

# 4 Experiments

## 4.1 Implementation Details

**Dataset.** In the experiments, we mainly evaluate our model on the NeRF Synthetic (27) dataset. The NeRF synthetic dataset consists of 8 path-traced scenes with multi-view images which we reconstruct into our textural mesh-based representation via nvdiffrec (29). Besides, we adopt 3D objects from Keenan's 3D model repository.

**Experiment Setup.** We optimize the 3D model on one RTX A6000 GPU with 48G memory. The optimization procedure lasts about average 500 iterations with 8 minutes for each 3D model. We use the Adam optimizer for both the texture and the environment map with an initial learning rate of 0.01 which gradually decreases to 1/10 every 5k iterations during the training process.

## 4.2 Comparison with Baseline

**Qualitative Comparison.** We compare ITEM3D with the optimization method based on the SDS loss. Specifically, Fig. 3 shows the results of editing texture and rendered image with the guidance of text prompts. While SDS-based method could edit textures along the direction of text prompt, their rendered images show the unrealistic appearance, sometimes overfitting to the text. In contrast,

Table 1: **Quantitative Comparisons**. We report two CLIP-based scores, *i.e.*, global score and directional score to evaluate the semantic quality of rendered images. '-' indicates not available. Our ITEM3D achieves better results than the SDS-based method. Besides, the inferiority of the performance without direction adjustment also shows the effect of this designed component.

| Method | Origin (**Ref.**) | ITEM3D | SDS-based | w/o dir. adjustment |
|---|---|---|---|---|
| Global Score↑ | 0.31 | **0.32** | 0.30 | 0.30 |
| Directional Score↑ | - | **0.23** | 0.18 | 0.16 |

Table 2: **User study** conducted with 33 participants. Each participant scores based on two evaluation criteria, *i.e.*, photorealism and text consistency. The range of scores is from 1 to 5, where 1 represents worst and 5 represents best.

| Method | Origin (**Ref.**) | ITEM3D | SDS-based |
|---|---|---|---|
| Photorealism ↑ | 4.18 | 3.77 | 2.77 |
| Text Consistency ↑ | - | 4.11 | 2.45 |

the texture edited by our ITEM3D can render realistic images with high quality, while remaining consistent with the input text prompt. The comparison indicates the effectiveness of the introduced relative direction of optimization and further direction adjustment. Besides, it can be noticed that our methods support segmentation-aware editing. Although the diffusion model lacks the capacity of recognizing the semantics in the texture map, it enables to edit the specific part of texture corresponding to a text prompt describing partial change. For example, with the prompt "A ficus with blue pot", the change in the texture precisely reflects to the part of the pot in the rendered images. It proves that the gradients can accurately back-propagate to the corresponding parts of the texture map via the rendered images.

**Quantitative Comparison.** Moreover, we conduct a quantitative comparison in the Tab. 1. To evaluate the semantic consistency, we choose objects from Keenan's 3D Model Repository, render their $512 \times 512$ RGB images after texture editing, and further compute the CLIP-Score of the rendered image and corresponding target text. CLIP-score contains two parts, *i.e.*, global score and directional score. Global score measures the similarity between the target text and the editing images, and directional score measures the similarity between two editing directions of text prompts and images, which are expressed as which can be presented as,

$$\text{Score}_{\text{global}} = \frac{T_{\text{tgt}} \cdot I_{\text{tgt}}}{\|T_{\text{tgt}}\|\|I_{\text{tgt}}\|}, \quad \text{Score}_{\text{direction}} = \frac{\Delta T \cdot \Delta I}{\|\Delta T\|\|\Delta T\|}, \tag{15}$$

where $T_{\text{tgt}}$ and $I_{\text{tgt}}$ are the embedding of target text and edited image encoded by the CLIP encoder, and $\Delta T$ and $\Delta I$ are expressed as,

$$\Delta T = T_{\text{tgt}} - T_{\text{src}}, \quad \Delta I = I_{\text{tgt}} - I_{\text{src}}. \tag{16}$$

As illustrated in Tab. 1, our method achieves better results than the SDS-based method.

**User Study.** Additionally, we perform a user study in Tab. 2 to further assess the quality of editing objects. Users are required to rate on a scale of 1 to 5, based on the following questions: (1) Are the edited objects realistic and natural (Photorealism)? (2) Are the edited objects accurately reflect the target text's semantics (Text Consistency)? As presented in Tab. 2, the results demonstrate the superior quality with higher realism and more text consistency of our proposed method as compared to the baselines.

### 4.3 Direction Adjustment

In this section, we further study the necessity of direction adjustment. We perform the ablation study in Fig. 4. Without the adjustment for the relative optimization direction, the texture shows a wired change that the duck gradually generates two heads and the color seems partially yellow and partially red. When applying the gradual adjustment, the duck bypasses the unnatural change and smoothly achieves the target appearance. The example of cattle shows a similar trend. In this experiment, it can be noticed that there exists unbalanced optimization for different parts of the

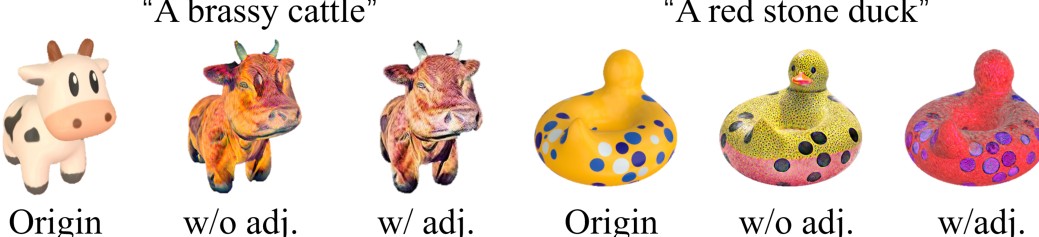

"A brassy cattle"    "A red stone duck"

Origin    w/o adj.    w/ adj.    Origin    w/o adj.    w/adj.

Figure 4: **Ablation study** of direction adjustment. The results without adjustment show a wired appearance, *i.e.*, dual heads and quadruple eyes. When applying gradual adjustment, the unrealistic artifacts are released, in result of natural appearance.

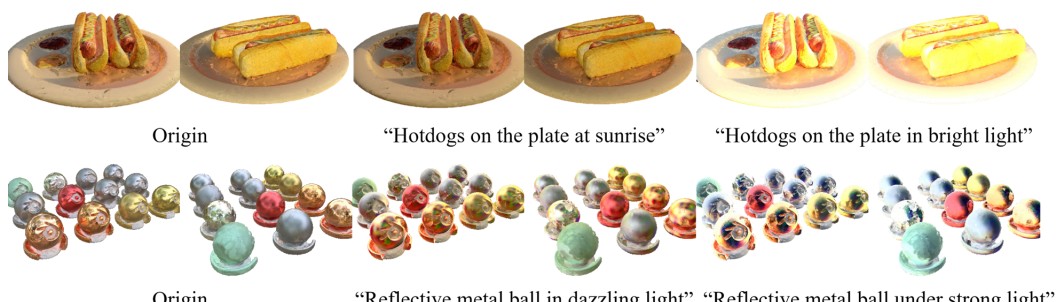

Origin                "Hotdogs on the plate at sunrise"        "Hotdogs on the plate in bright light"

Origin         "Reflective metal ball in dazzling light"  "Reflective metal ball under strong light"

Figure 5: **Relighting** results under the condition of an illumination-aware text prompt. Keeping the texture constant, ITEM3D has capacity of explicit control over the lighting under the guidance of prompt related to the environment map.

texture. The optimization scheme of simple pieces of texture converges quickly, while more complex modifications require longer time, which in turn over-tunes easy parts leading to poor results. We also compute the two CLIP score for the results without direction adjustment in Tab. 1. It shows that the adjustment indeed helps to maintain the major semantics.

### 4.4 Illumination-aware Editing

The disentangled representation of environment map empowers ITEM3D to explicitly control the lighting under the guidance of a text prompt aiming to relight the 3D model. The results of illumination-aware editing are demonstrated in Fig. 5. As shown, given the prompt including lighting information such as "sunrise", "bright light", and "dazzling light", ITEM3D enables to edit the environment map along the direction led by the prompt. It is valuable to prove that the lighting condition of a 3D model can be learned solely from the text through the bridge of rendered 2D images.

## 5 Conclusion and Limitations

In conclusion, our ITEM3D model presents an efficient solution to the challenging task of texture editing for 3D models. By leveraging the power of diffusion models, ITEM3D is capable to optimize the texture and environment map under the guidance of text prompts. To address the semantic ambiguity between text prompts and images, we replace the traditional score distillation sampling (SDS) with a relative editing direction. We further propose a gradual direction adjustment during the optimization procedure, solving the unbalanced optimization in the texture.

Despite the promising editing results, our ITEM3D still remains several limitations which should be solved in future work. The major limitation is that there remains irremovable noise in some samples. Because of the synthesis mechanism of the diffusion model, our ITEM3D extremely depends on the denoising ability of the pre-trained U-Net. Another limitation is that the adjustment by the source description is non-essential. Our further work aims to explore the learning scheme to solve the problem of unbalanced optimization in the texture.

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
