# OpenReview forum: "ITEM3D: Illumination-Aware Directional Texture Editing for 3D Models"
_NeurIPS.cc/2023/Conference — Submitted to NeurIPS 2023_

### Official Review · Reviewer_CxP2 · 2023-06-26

**Soundness:** 2 fair
**Presentation:** 2 fair
**Contribution:** 2 fair
**Rating:** 4
**Confidence:** 5

**Summary:**

The research introduces ITEM3D, a model that enhances texture editing in 3D modeling, addressing challenges like complexity and text ambiguity. Leveraging a diffusion model, ITEM3D uses images to bridge text and 3D representations, optimizing texture and environment map. It employs a relative editing direction, reducing noise difference and semantic ambiguity between source and target texts, and adjusts direction during optimization to limit texture domain deviation. Experiments show ITEM3D outperforms previous methods and can effectively control lighting.


**Strengths:**

+ The research introduces an optimization pipeline for texture and environment map editing in 3D models, adhering to text prompts.
+ This paper presents a relative direction diffusion-based approach for 3D texture optimization, mitigating issues of noisy details and inconsistent appearances resulting from semantic ambiguity between texts and images.


**Weaknesses:**

1. This paper asserts its realism and efficiency in generating new textures through optimization, yet its claims are evaluated solely on rendered images. It remains uncertain whether this method is applicable to in-the-wild objects. I suggest that the authors conduct additional experiments using real 3D objects, such as those in the DTU dataset. Furthermore, I find a lack of supportive material regarding its efficiency claims. Given that the method works based on optimization, I question whether it can truly generate high-quality textures swiftly, as stated in the introduction.

2. The introduction lacks clarity and adequate context. The paper outlines two challenges when applying the diffusion model to 3D objects, but it doesn't discuss any related works aimed at addressing these challenges. This omission makes it difficult to gauge the novelty of the proposed method in relation to existing solutions.

3. The proposed relative direction loss appears strikingly similar to that of NeRF-Art[1], albeit the latter employs the CLIP model to implement this constraint. The authors should highlight the distinguishing factors between their method and NeRF-Art in the introduction and add a direct comparison.

	[1] Wang, C., Jiang, R., Chai, M., He, M., Chen, D. and Liao, J., 2023. Nerf-art: Text-driven neural radiance fields stylization. IEEE Transactions on Visualization and Computer Graphics.

4. In Figure 4, the examples provided are not representative enough. It would be beneficial to showcase results with brown, golden, and porcelain materials. The current examples make it difficult to evaluate the method's performance.

**Questions:**

1. The reconstruction results presented in Figure 5 seem inferior to those achieved by the original NeRF. The results appear quite coarse. Could the authors elaborate on the reasons for this?

**Limitations:**

This paper has acknowledged its limitations.

---

> ### Author Rebuttal · Authors · 2023-08-09
>
> We would like to express our sincere gratitude for your valuable feedback on our paper.
>
> **Q1: [Experiments on real-world dataset and experiments about efficiency]**
>
> **A1**: We have incorporated a qualitative experiment on a real-world 3D dataset, as shown in **Fig.1** of the rebuttal PDF. The dataset comprises hundreds of products listed in E-commerce, and each object is reconstructed from 300 multi-view images captured by a professional camera.
>
> We present three examples in the dataset: a vegetable cat, a piggy doll, and a red sneaker. As demonstrated in **Fig.1** of the rebuttal PDF, our model, ITEM3D, successfully synthesizes realistic and natural textures based on the given text guidance. For instance, when provided with the prompt "a vegetable tiger toy", ITEM3D edits the original object into a cute toy tiger with detailed fur and vegetable decorations. When given the prompt "golden sneaker," ITEM3D can bake the golden material instead of original material into the texture and show a realistic shoe. Similarly, the example of "a pink porcelain piggy toy" showcases ITEM3D's capacity for appearance and material editing.
>
> To support our efficiency claims, we have conducted experiments to compare the efficiency of our approach with the state-of-the-art method, instruct-NeRF2NeRF. In Table below, we present the results of comparing editing time and memory consumption. Our findings demonstrate that ITEM3D outperforms instruct-NeRF2NeRF, requiring significantly less time (50 times less) while maintaining comparable memory consumption during texture editing.
>
> |    method     | Instruct NeRF2NeRF | ITEM3D |
> | :-----------: | :----------------: | :----: |
> | training time |        10h         | 10min  |
> |  GPU memory   |        15GB        |  9GB  |
>
> Thank you for your valuable input, and we appreciate your emphasis on both qualitative results and efficiency comparisons. It helps strengthen our claims and provides a more comprehensive evaluation of our approach.
>
> **Q2: [The introduction lacks clarity and adequate context]**
>
> **A2**: We appreciate your attention to the lack of clarity and adequate context in the introduction. We will ensure appropriate revisions are made in the final version. Previous generative models, such as DreamFusion, faced the challenge of injecting 2D priors into 3D objects. This resulted in occasional failure to generate accurate 3D shapes and corresponding textures due to insufficient 3D knowledge. Additionally, the direct utilization of SDS loss, as proposed in DreamFusion, mislead the object's starting point, leading to a blurry image. Alternative works, such as Instruct NeRF2NeRF and HeadSculpt, utilized InstructPix2Pix to provide relative direction. However, our ITEM3D propose to obtain relative direction, rather than directly modifying the prompt. Our method yields comparable results, as shown in **Fig.2** of the rebuttal PDF.
>
> **Q3: [The proposed relative direction loss is similar to that of NeRF-Art]**
>
> **A3**: As mentioned in line 206 of our study, our method draws inspiration from the idea presented in StyleGAN-Nada. StyleGAN-Nada first propose the concept of directional loss to the CLIP model, and NeRF-Art is one of the approaches that follows this direction for CLIP-based editing tasks.
>
> However, applying the directional CLIP loss directly to diffusion models is not a straightforward task. In our research, we have explored the similarities and differences between CLIP-based editing and diffusion-based editing. By understanding these distinctions, we were able to make improvements based on the specific characteristics of the SDS loss. This enabled us to successfully leverage the advantages of the directional loss and apply them to texture editing based on the diffusion prior. By adapting and refining the directional loss for diffusion models, we aimed to enhance the texture editing capabilities and achieve more realistic and accurate results.
>
> Besides, we compare our method with CLIP-based model in **Fig.4** of the rebuttal PDF, to demonstrate the advantages of diffusion-based models.
>
> **Q4: [Results with brown, golden, and porcelain materials]**
>
> **A4**: We present additional editing results on real-world objects in **Fig.1** of the rebuttal PDF. Our demonstrations showcase a remarkable proficiency in object editing, exemplified by the conversion of an original cat toy into a vegetable tiger toy, a piggy doll into a pink porcelain piggy toy and a sneaker into a golden sneaker. The edited cat toy exhibits a fantastic furry material, which is truly impressive. Similarly, the golden sneaker showcases the distinctive texture of a metallic golden material. Furthermore, the piggy doll demonstrates our capability in appearance and material editing through the use of the "pink porcelain piggy toy" prompt, resulting in a toy made of porcelain material.
>
> **Q5: [The reason for coarse results in Fig.5]**
>
> **A5**: In **Fig.5** of original paper, there are two main reasons that contribute to the coarse results observed.
> The first reason is the environment lighting condition. In some cases, when the lighting conditions are strongly bright, it can obscure or hide certain texture details. This can result in the coarser appearance of the edited object.
>
> The second reason is related to the use of nvdiffrec for efficient differential rendering. While nvdiffrec enables faster rendering compared to the lengthy optimization process of the original NeRF, the learning process of nvdiffrec may occasionally lead to inferior results. This may affect the overall quality and fine details of the geometry in certain cases.
>
> It is important to note that these limitations are inherent to the specific techniques employed in our study. Future advancements in differentiable rendering may help address these challenges and improve the quality of texture editing results.
> We appreciate your understanding and acknowledgment of these limitations.

---

> > ### Comment · Reviewer_CxP2 · 2023-08-21
> >
> > The authors have addressed most of my concerns, including the loss function, results with various materials such as brown, golden, and porcelain, and experiments conducted on a real-world dataset. However, I continue to have the following concerns:
> >
> > (1) The comparison of relative direction loss with NeRF-Art should be introduced and discussed in the introductory section.
> >
> > (2) The efficiency comparison as it currently stands is not equitable, since different 3D representations are employed in Instruct-Nerf2Nerf and the proposed method. This discrepancy is likely the reason why the paper demonstrates much faster results than Instruct-Nerf2Nerf. The authors should compare both methods under the same 3D representation conditions and provide a thorough analysis to explain why this method is more efficient.
> >
> > (3) The quality of the coarse results presented in Fig.5 appears to be lacking. I do not believe that the lighting conditions are to blame. In contrast, other works such as NeRF-Factor seem to handle similar cases with more competence. It seems that the reconstruction itself may be the issue. I would like to understand why this work is unable to effectively address this specific case.

---

> > > ### Author Response · Authors · 2023-08-21
> > >
> > > Thank you for your insightful comments and suggestions regarding our paper. We have carefully considered your feedback. Below is our response:
> > >
> > > **A1**: It is better to include a discussion on the relative loss in the introduction to elucidate our contribution towards enhancing the SDS loss. We will discuss the first proposed relative clip loss in StyleGAN-Nada and explore related works such as NeRF-Art, and others.
> > >
> > > **A2**: Indeed, one of the primary factors contributing to the faster speed of our method compared to Instruct NeRF2NeRF is the utilization of DMTET instead of NeRF. However, unlike Instruct NeRF2NeRF, our approach circumvents the need for iterative dataset modifications to maintain multi-view consistency when editing objects. By directly editing the disentangled texture map, our method preserves the inherent multi-view consistency and avoids the alternative updating of 3D representation and data, resulting in time savings. On the other hand, it is true that ITEM3D exhibits lower efficiency compared to end-to-end architectures due to the optimization process involved. However, end-to-end architectures typically require a large volume of 3D datasets and are often limited to class-specific tasks, such as Rodin [1].
> > >
> > > **A3**: We acknowledge that the reconstruction method we adopted, Nvdiffrec [5], does have certain limitations. The challenge of reconstructing surfaces with high specular reflectance has been extensively discussed in a recent work called ENVIDR [2]. When faced with this challenge, most existing methods take one of two major approaches.
> > >
> > > The first category of methods, such as the original NeRF and its extensions [3, 4], involves explicitly representing virtual lights or images underneath the surface to capture complex view-dependent appearance. While this approach can improve rendering quality, it often sacrifices the accuracy of the reconstructed surface and limits the ability to edit scenes.
> > >
> > > Nvdiffrec adopted by our method falls into the second category, which incorporates knowledge of inverse rendering to model the interaction between light and surface. These methods [5, 6, 7] decompose rendering parameters, but they often suffer from relatively lower rendering quality compared to top-performing NeRF models. This is because that the simplified or approximated rendering equation used in these models cannot account for all complex rendering effects.
> > >
> > > While NeRF-factor [7] also decomposes rendering parameters, it builds upon a pre-trained NeRF and employs additional joint optimization to further enhance the quality. So, NeRF-factor can handle such cases with more competence. However, the complex procedure of NeRF-factor results in a runtime of nearly 8 hours. Even excluding the base NeRF reconstruction, the decomposition and joint optimization process alone requires approximately 50 minutes.
> > >
> > > To address these challenges and improve the quality of reconstruction results, we will explore advancements such as ENVIDR [2] in our future work. ENVIDR learns an approximation of physically based rendering (PBR) using three decomposed MLPs, which are trained using images with various materials and environments synthesized by existing PBR renderers.
> > > This may help to mitigate the limitations we have encountered.
> > >
> > > [1] Wang T, Zhang B, Zhang T, et al. Rodin: A generative model for sculpting 3d digital avatars using diffusion[C]//Proceedings of the IEEE/CVF Conference on Computer Vision and Pattern Recognition. 2023: 4563-4573.
> > >
> > > [2] Liang R, Chen H, Li C, et al. ENVIDR: Implicit Differentiable Renderer with Neural Environment Lighting[J]. arXiv preprint arXiv:2303.13022, 2023.
> > >
> > > [3] Liu L, Gu J, Zaw Lin K, et al. Neural sparse voxel fields[J]. Advances in Neural Information Processing Systems, 2020, 33: 15651-15663.
> > >
> > > [4] Yu A, Li R, Tancik M, et al. Plenoctrees for real-time rendering of neural radiance fields[C]//Proceedings of the IEEE/CVF International Conference on Computer Vision. 2021: 5752-5761.
> > >
> > > [5] Munkberg J, Hasselgren J, Shen T, et al. Extracting triangular 3d models, materials, and lighting from images[C]//Proceedings of the IEEE/CVF Conference on Computer Vision and Pattern Recognition. 2022: 8280-8290.
> > >
> > > [6] Zhang K, Luan F, Wang Q, et al. Physg: Inverse rendering with spherical gaussians for physics-based material editing and relighting[C]//Proceedings of the IEEE/CVF Conference on Computer Vision and Pattern Recognition. 2021: 5453-5462.
> > >
> > > [7] Zhang X, Srinivasan P P, Deng B, et al. Nerfactor: Neural factorization of shape and reflectance under an unknown illumination[J]. ACM Transactions on Graphics (ToG), 2021, 40(6): 1-18.

---

> ### Comment · Area_Chair_7gK8 · 2023-08-18
>
> Reviewer CxP2,
>
> Please read the rebuttal provided by authors and raise a discussion if your concerns are not well addressed.
>
> Best,
> AC

---

### Official Review · Reviewer_1P22 · 2023-07-06

**Soundness:** 2 fair
**Presentation:** 2 fair
**Contribution:** 3 good
**Rating:** 7
**Confidence:** 5

**Summary:**

Given a set of multi-view calibrated images, the authors aim to first reconstruct the 3D objects based on the input images and then edit them via a pre-trained diffusion model. To accomplish this, they developed ITEM3D, which utilizes a differentiable marching tetrahedron (DMTet) as its 3D implicit representation. To achieve the editing task, the authors proposed two methods: (1) relative direction based optimization that respects both descriptive and editive prompts, and (2) a gradient direction adjustment that integrates changes from previous iterations to improve network robustness. Qualitative and quantitative experiments demonstrated ITEM3D's superior capabilities compared to current state-of-the-art methods.

**Strengths:**

The strengths of this paper can be summarized as:
1. The paper employs both descriptive and editive directions to effectively address the editing task. Although it's adapted from the recent DDS paper, I believe the tricks proposed by the authors are useful.
2. Additionally, the authors integrate changes to adjust the gradient direction, enhancing the network's robustness.

**Weaknesses:**

The weaknesses of this paper can be summarized as:
1. The paper overlooks several noteworthy recent studies, such as Make-it-3D, DreamBooth3D, DreamAvatar, Instruct-NeRF2NeRF, Rodin, and so on, which should be included in the references.
2. The authors fail to provide a thorough discussion and qualitative comparisons with Instruct-NeRF2NeRF, the most recent and relevant paper.
3. the ablation study presented in Figure 4 may not be immediately apparent. The "w/o adj" may be a better result for the "A brassy cattle" scenario.

**Questions:**

Besides the weaknesses listed above, I have several questions for the authors to consider in the rebuttal:
1. The descriptive prompt may not perfectly describe the object since the 3D DMTet is initially optimized via the input multi-view images, which could potentially impact the final results in a negative manner.
2. Following the approach proposed in HeadSculpt, it may be beneficial to input the rendered images from the original DMTet instead of the optimized DMTet. I am wondering whether the IESD loss proposed in HeadSculpt would achieve comparable or superior results compared to the approaches in ITEM3D
3. Figure 2 currently depicts prompts as "A green chair" and "A green wooden chair." It may be worth exploring the possibility of changing the prompts to "A green chair" and "make it wooden," as proposed in HeadSculpt.

**Limitations:**

Limitations and potential impacts have been discussed in the submission.

---

> ### Author Rebuttal · Authors · 2023-08-09
>
> Thank you sincerely for your positive and encouraging feedback on our paper.
>
> **Q1: [Lack of related works]**
>
> **A1**: We appreciate your suggestion to include these references and provide a more comprehensive discussion of the related works.
> Based on recent state-of-the-art studies, we have investigated and briefly discussed the relevant works here.
>
> Among these recent studies, Make-it-3D, DreamBooth3D, and Instruct-NeRF2NeRF are more closely related to our methods as they also utilize text prompts to guide the appearance editing of 3D objects. These studies employ NeRF as the 3D representation, whereas ITEM3D utilizes a disentangled representation consisting of mesh, texture, and environment map.
>
> Instruct-NeRF2NeRF and DreamBooth3D both utilize 2D diffusion to update the training images and fine-tune the NeRF training. They also design updating strategies to maintain multi-view consistency during the editing process. Make-it-3D, following Dreamfusion, incorporates the SDS loss to directly optimize the NeRF and further refines the results using a texture point cloud. In contrast, ITEM3D chooses to optimize the texture map directly while preserving consistency due to its disentangled representation.
>
> Other works, such as Rodin, DreamAvatar, and AvatarBooth, focus on diffusion-based text-driven avatar synthesis. Rodin trains a tri-plane based diffusion model using approximately 100k 3D head models. DreamAvatar and AvatarBooth optimize a NeRF-like representation using the SDS loss, similar to DreamFusion and Make-it-3D. These methods leverage SMPL to initialize a parameterized body, which is used in the body generation process. In contrast, our method focuses on generalized editing, and we aim to explore our capability in avatar editing in future work.
>
> Your input helps enhance the context and understanding of our research within the broader field of text-guided 3D object editing and synthesis. We will add the discussion above in our final version.
>
> **Q2: [Qualitative comparisons with Instruct-NeRF2NeRF]**
>
> **A2**: We provide qualitative comparisons with Instruct-NeRF2NeRF in **Fig.2** of the rebuttal PDF. Both Instruct-NeRF2NeRF and ITEM3D are capable of achieving text-consistent texture editing. However, Instruct-NeRF2NeRF falls behind our method in certain aspects.
>
> For example, when editing the chair using prompts such as "Turn the chair into a red stone chair" or "Turn the chair into a green wooden chair", Instruct-NeRF2NeRF exhibits a lack of details and presents a smooth material. In contrast, ITEM3D's edited textures showcase clear patterns and wood grain, resulting in a more realistic appearance.
>
> Furthermore, the ficus case highlights another limitation of Instruct-NeRF2NeRF in handling disentangled editing. When prompted to "Turn the pot into a blue pot," Instruct-NeRF2NeRF applies the blue color to both the pot and the ficus, failing to disentangle the two components. In contrast, our method, ITEM3D, successfully disentangles the pot and accurately turns it blue while leaving the ficus unchanged.
>
> These qualitative comparisons demonstrate that ITEM3D outperforms Instruct-NeRF2NeRF in terms of capturing fine details, maintaining texture patterns, and effectively handling disentangled editing tasks.
>
> **Q3: [The ablation study presented in Fig.4 of original paper]**
>
> **A3**: In the example you mentioned, the result labeled as "w/o adj" in our figure fails to eliminate the original two cute eyes and instead shows four eyes on the cattle head. We acknowledge that this misrepresentation could be overlooked due to visual deception.
>
> We sincerely apologize for any confusion or misleading caused by this figure. To rectify the situation, we have revised the figure and marked the problematic areas with red circles to draw attention to the issue. This clarification aims to ensure transparency and accuracy in our presentation. We appreciate your understanding and diligence in pointing out this discrepancy.
>
> **Q4-Q5: [Ablation study on IESD loss]**
>
> **A4-A5**: HeadSculpt is a recent arxiv paper submitted in June. We recently investigated the method proposed in the HeadSculpt paper and introduced its IESD (Identity-aware editing score distillation) loss into our ablation study, as shown in **Fig.1** of the rebuttal PDF. The IESD loss utilized in HeadSculpt aims to balance the source object and target object, presented as $\alpha\left(\epsilon_{\phi}\left(z_{t};\hat{y},t\right)-\epsilon\right)+\beta\left(\epsilon_{\phi}\left(z_{t};y,t\right)-\epsilon\right)$, where in HeadSculpt, $\alpha$ and $\beta$ are set at 0.6 and 0.4, respectively. In contrast, our method employs $\alpha$ and $\beta$ values of 1 and -1, respectively, with both methods sharing a unified mathematical form.
>
> As depicted in **Fig.1** of the rebuttal PDF, both our loss function and the IESD loss function yield comparable results. They are capable of editing the texture into another realistic and natural texture based on the given prompt, with minimal differences between the outcomes. It need further research to determine the superiority of these two different parameter settings.
>
> **Q6: [Changing the prompts to "make it wooden"]**
>
> **A6**: We have conducted an ablation study, as shown in **Fig.5** of the rebuttal PDF, to explore the effect of different text descriptions.
> As shown, using prompt "make it wooden" in ITEM3D cause degradation  in the editing results.
>
> This is because that our method introduces a relative direction for editing, which makes it more appropriate to use source and target text prompts as starting and ending points. On the other hand, prompt formats that directly describe the direction, such as "make it wooden", may be more suitable for models like instruct-pix2pix, which is the base model of HeadSculpt and instruct-NeRF2NeRF.

---

> > ### Comment · Reviewer_1P22 · 2023-08-11
> >
> > Post Rebuttal:
> >
> > Thanks for the efforts from the authors. With regards to my preliminary reviews and the authors' rebuttal:
> >
> > A1. I kind of agree with the authors.  But I believe it would be beneficial to reference and incorporate discussions involving similar existing methods, even if they do not address the exact same task. I encourage the authors to conduct a thorough literature review to ensure that no relevant papers have been overlooked.
> >
> > A2. I believe the comparisons with Instruct-NeRF2NeRF are evident, even though determining superiority becomes challenging since it relies more on personal evaluation rather than quantitative evaluation.
> >
> > A3. But I think the results with two cute eyes are better as we may want to keep the original identity, right?
> >
> > A4-A5. Thanks for the experiments.
> >
> > A6. Thanks for the ablation studies. Then I think it's better to solve this problem. Because in real-world applications, we often lack a specific and definitive text prompt for the subject, such as "A green chair."
> >
> > Overall I tend to keep or upgrade the original rating. Thanks.

---

> > > ### Author Response · Authors · 2023-08-11
> > >
> > > Thank you for your positive feedback. We appreciate your acknowledgment of the value of our experiments.
> > >
> > > **A1**:
> > >  We sincerely acknowledge your point about the importance of thoroughly reviewing the existing literature. While they may not focus on the exact same task, we agree that they can contribute significantly to our work. Therefore, we have recently conducted an exhaustive review of recent works in this area. The revised version will include a thorough discussion of these works, enabling us to situate our research in the broader context and highlight its unique contributions.
> > >
> > >  **A3**:
> > > To some extent, you are correct. We have taken your suggestion into careful consideration regarding the target of our editing. It has come to our attention that preserving the identity holds value in specific downstream tasks, meanwhile it should not sacrifice the reasonableness of results. In this case, the ranking of the results is as follows: two cute eyes > two real eyes (the result with adjustments) > four eyes (the erroneous result without adjustments). Consequently, we should make an effort to prioritize both identity preservation and the prevention of erroneous outcomes in future similar cases.
> > >
> > > **A6**:
> > > I agree that the subject lacks a definitive description. In fact, our method does not necessarily require an exact prompt. The source and target prompts serve the purpose of determining the relative direction of the editing process. Even a general and coarse description can have an impact on complex objects. Our main objective is to express our intention as the distinction between the source and target texts, while ensuring that the prompts remain reasonable.
> > >
> > > However, it is indeed more convenient to directly express the editing purpose like "make it wooden".  We will explore the possibility to improve our methods on this aspect.

---

> > > > ### Comment · Reviewer_1P22 · 2023-08-12
> > > >
> > > > Thanks. Good luck. I am also looking forward to the release of your revised paper, which incorporates the rebuttal materials and the promises to all reviewers.

---

> > > > > ### Author Response · Authors · 2023-08-13
> > > > >
> > > > > We sincerely thank you for providing a novel perspective to our research and valuable insights. We will incorporate these considerations into the final revised paper to our best effort.

---

### Official Review · Reviewer_67rK · 2023-07-07

**Soundness:** 2 fair
**Presentation:** 3 good
**Contribution:** 2 fair
**Rating:** 3
**Confidence:** 4

**Summary:**

This paper proposes ITEM3D, a model for automatic 3D object editing according to text prompts. ITEM3D bridges the gap between 3D representation and natural images using rendered images. It optimizes disentangled texture and environment maps using a relative editing direction to bypass ambiguous text descriptions. It gradually adjusts the editing direction to reduce deviation caused by texture projection. Results demonstrate ITEM3D outperforms SDS-based methods on various 3D objects. It also allows for explicit control over lighting with text-guided relighting. This paper also suggests directly applying the diffusion model to 3D objects is challenging due to conflicts in optimization goals and extreme semantic bias.

**Strengths:**

Optimizing disentangled texture and environment maps using a relative editing direction to bypass ambiguous text descriptions looks interesting. Results demonstrate ITEM3D outperforms SDS-based methods on various 3D objects. The explicit control over lighting with text also looks interesting.

**Weaknesses:**

One major weakness of the proposed method is the lack of comparisons with other important related works, such as CLIPNeRF, ARF[1], and SINE[2]. While CLIPNeRF is cited, the others are not. Additionally, there are several related works that do not use text descriptions for editing NeRF, including NeRV[3], NeRD[4], NerFactor[5], and EditNeRF[6], that are also missing from the comparison. Another related line of work, SDFusion[7], is not discussed at all. Although the authors may have overlooked recent related works in the crowded field of text-based editing with NeRF + Diffusion models, it is crucial to compare the proposed method to CLIPNeRF and ARF to understand its advantages. Comparisons to intrinsic image decomposition-based methods are also important, especially for relighting, to determine the strengths and weaknesses of the proposed method. While the expectation is not for the method to outperform NeRD or NeRFactor, comparing it to these methods is important to identify what needs to be done to achieve similar results.

Another important and simple baseline is missing: how do the results compare to a simple 2D-based image editing method without using explicit 3D texture and illumination map optimization?

[1] Zhang, Kai, et al. "Arf: Artistic radiance fields." European Conference on Computer Vision. Cham: Springer Nature Switzerland, 2022. \
[2] Bao, Chong, et al. "Sine: Semantic-driven image-based nerf editing with prior-guided editing field." Proceedings of the IEEE/CVF Conference on Computer Vision and Pattern Recognition. 2023. \
[3] Srinivasan, Pratul P., et al. "Nerv: Neural reflectance and visibility fields for relighting and view synthesis." Proceedings of the IEEE/CVF Conference on Computer Vision and Pattern Recognition. 2021. \
[4] Boss, Mark, et al. "Nerd: Neural reflectance decomposition from image collections." Proceedings of the IEEE/CVF International Conference on Computer Vision. 2021.  \
[5] Zhang, Xiuming, et al. "Nerfactor: Neural factorization of shape and reflectance under an unknown illumination." ACM Transactions on Graphics (ToG) 40.6 (2021): 1-18. \
[6] Liu, Steven, et al. "Editing conditional radiance fields." Proceedings of the IEEE/CVF international conference on computer vision. 2021. \
[7] Cheng, Yen-Chi, et al. "Sdfusion: Multimodal 3d shape completion, reconstruction, and generation." Proceedings of the IEEE/CVF Conference on Computer Vision and Pattern Recognition. 2023.


**Questions:**

It would be helpful to compare the current methods with other recent works mentioned in the weakness to better understand their effectiveness. Additionally, it is unclear how text-guided lighting is mapped to the disentangled illumination representation. Is it solely through the use of diffusion priors or are new text embeddings or directions learned for specific lighting conditions? If diffusion models already include such priors, would optimization in 2D suffice without the need for explicit 3D texture and illumination map optimization?

**Limitations:**

The paper lacks several key related works and comparisons to them, including simple 2D-based comparisons.

---

> ### Author Rebuttal · Authors · 2023-08-09
>
> Thank you for providing additional references and addressing the lack of comparison with other baselines in our paper.
>
> **Q1-Q2: [Lack of related works and comparison experiments]**
>
> **A1-A2**: CLIP-NeRF, ARF, and SINE are CLIP-based editing models that, compared to diffusion-based models, lack the capability to handle complex editing tasks in generalized objects.To demonstrate the advantages of NeRF + Diffusion models, we have compared our ITEM3D approach with two baselines: CLIP-NeRF and a 2D baseline called ControlNet. In **Fig.4** of our rebuttal PDF, we present the results of this comparison.
>
> For the CLIP-NeRF baseline, we utilized the official code and the provided pre-trained lego NeRF model from the CLIP-NeRF repository to reproduce its performance. Given the prompt "A red real excavator", we observed that the 2D baseline successfully edited the input views into a realistic red excavator. However, the results lacked multi-view consistency, with distinct patterns and slight variations in the excavator's color across the collection of five-view images. On the other hand, while CLIP-NeRF achieved 3D consistency, it failed to edit the excavator into a red color and instead changed the color of the base to red. In contrast, our ITEM3D approach was able to both edit the color according to the prompt and maintain the inherent 3D consistency of the object.
>
> Besides, In **Fig.2** of our rebuttal PDF, we present the results of the comparison experiment between ITEM3D and the recent state-of-the-art work, Instruct-NeRF2NeRF. Both ITEM3D and Instruct-NeRF2NeRF are capable of achieving text-consistent editing results with natural textures. However, there are certain areas where Instruct-NeRF2NeRF falls short compared to our approach.
> For example, when given the prompt "Turn the pot into a blue pot", Instruct-NeRF2NeRF fails to disentangle the pot and the ficus, resulting in undesired changes to the ficus. On the other hand, our ITEM3D approach is able to successfully edit the pot into a blue color while preserving the integrity of the ficus. When editing chairs, Instruct-NeRF2NeRF tends to synthesize smooth textures with little details. In contrast, our ITEM3D approach performs better in capturing and preserving fine details during texture editing.
>
> Similar to Instruct-NeRF2NeRF, SDFusion optimizes the neural radiance fields (NeRF) to edit the texture of reconstructed 3D object. The optimized process is guided by the sds loss as proposed in DreamFusion.
>
> For non-text editing works, NeRV enables the rendering of novel views with varying lighting conditions, including indirect illumination effects. NeRD decomposes a scene into its shape, reflectance, and illumination components, enabling real-time rendering with novel illuminations. NeRFactor introduces a method to reconstruct the shape and spatially-varying reflectance of an object from multi-view images, allowing for rendering novel views under different lighting conditions and material editing. These methods utilize decomposed representations to edit illumination and material aspects, which aligns with the approach of our methods. However, the main contribution of our work lies in text-guided texture editing, which is an aspect they do not specifically focus on.
>
> **Q3: [How to map text-guided lighting to the disentangled illumination representation]**
>
> **A3**: In our pipeline, we employ a process that disentangles the 3D model into three components: the appearance texture map, the environment lighting map, and the geometric mesh. This disentanglement allows us to edit each component separately to achieve the desired changes.
>
> During the illumination editing stage, we fix the texture and mesh components and focus on optimizing the environment lighting map based on the given text prompt. By leveraging the differential rendering process, where the 3D model is rendered into 2D images, our diffusion model can utilize the lighting prior information from these 2D images to guide the optimization process for the lighting map.
>
> However, it's important to note that directly optimizing the 2D images using the diffusion model cannot maintain multi-view consistency. This limitation is overcome by our method, which preserves the multi-view consistency by utilizing the inherent 3D representation of the object. By disentangling the 3D model and leveraging the differential rendering process, our approach maintains the 3D consistency of the object while achieving desired edits. This ensures that changes made to the lighting map, texture map, or geometric mesh are coherent across different views and result in visually consistent and realistic edits.

---

> > ### Comment · Reviewer_67rK · 2023-08-16
> > **Response to author rebuttal**
> >
> > Thank you for taking the time to address the concerns raised in the review. I appreciate your efforts in providing clarifications. However, there are still some areas that require further elaboration and evidence. Please find my comments below:
> >
> > - **Comparison with ARF and SINE:** Thank you for the comparison to CLIP-NeRF (CVPR 2022). However, a comparison between the proposed approach, ARF (ECCV 2022), and SINE (CVPR 2023) would be valuable, especially since both have been shown to outperform CLIP-NeRF. The claim that these methods "lack the capability to handle complex editing tasks in generalized objects" needs experimental evidence. For instance, SINE has demonstrated strong editing capabilities on real cars, which have intricate material properties. I'd like evidence supporting that ITEM3D outperforms ARF and SINE.
> >
> > - **Related Work:** The distinction provided with NeRV, NeRD, and NeRFactor is insightful. I hope these papers will be incorporated into the related work section in any revised version of this paper.
> >
> > - **Text-guided Lighting:** Thank you for clarifying how text-guided lighting is mapped. However, the claim that "directly optimizing the 2D images using the diffusion model cannot maintain multi-view consistency" needs substantiation. Methods like SDFusion and Dreamfusion have shown strong multi-view consistency. It would be beneficial to see experimental evidence backing this claim.

---

> > > ### Author Response · Authors · 2023-08-17
> > >
> > > Thank you for your detailed and insightful feedback.
> > >
> > > * **Comparison with ARF and SINE**:
> > >
> > > Regarding the comparison with ARF and SINE, we apologize for the coarse discussion in our initial rebuttal. We would like to address them explicitly here and provide a more thorough discussion in our revised paper.
> > >
> > > ARF and SINE are both NeRF-based optimization methods that utilize image guidance. In contrast, our method focuses on text-guided texture editing. While ARF and SINE can be extended to text-prompt editing, they mainly rely on an off-the-shelf text-guided method called Text2live [1], as mentioned in the SINE paper. In the context of text-editing applications, SINE first utilizes Text2live to edit a single-view image corresponding to its pre-trained NeRF. The edited single-view image then serves as the input target image to optimize the NeRF representation, following the same procedure as the image-guided pipeline.
> > >
> > > There are several differences in the objectives of these methods. ARF aims at scene stylization, focusing on global color in the 3D scene but ignoring fine-grained details. On the other hand, SINE emphasizes detailed semantic editing.
> > >
> > > To summarize the differences among CLIP-NeRF, SINE, ARF, and our ITEM3D:
> > > |               | Guidance |    Task     |        Representation         |
> > > | :-----------: | :------: | :---------: | :---------------------------: |
> > > | **CLIP-NeRF** |   Text   |   Editing   |             NeRF              |
> > > |   **SINE**    |  Image   |   Editing   |             NeRF              |
> > > |    **ARF**    |  Image   | Stylization |             NeRF              |
> > > |  **ITEM3d**   |   Text   |   Editing   | Explicit texture and mesh |
> > >
> > > As shown, CLIP-NeRF and SINE are more closely aligned with our task, while ARF is not. We would have liked to conduct a comprehensive comparison with both CLIP-NeRF and SINE. However, since the code for SINE has not been released yet, it would be challenging for us to reproduce its results within the limited timeframe of this rebuttal. Therefore, we have provided a comparison with CLIP-NeRF in our current submission. Once the official code for SINE becomes available, we will perform the necessary comparison and include the results in our final version. Additionally, to demonstrate the effectiveness of our method in editing 3D objects using text, we conducted a comparative experiment with Instruct-NeRF2NeRF[2] in **Fig.2** of the rebuttal PDF to provide further support for ITEM3D. Because Instruct-NeRF2NeRF is also more related to our ITEM3D, with text guidance, diffusion-based editing task and NeRF representation.
> > >
> > > **Thank you once again for your valuable reference, and we ensure that all the points raised in the review will be incorporated in our revised paper.**
> > >
> > > * **Related Work**:
> > >
> > > We appreciate the acknowledgement. We will certainly incorporate these papers into the related work section to highlight their contributions and distinguish them from our own method.
> > >
> > > * **Text-guided Lighting**:
> > >
> > > Indeed, SDFusion and Dreamfusion are capable of preserving 3D consistency; however, they do not directly optimize 2D images but instead optimize the implicit 3D representation. By "directly optimizing 2D images", we refer to the utilization of diffusion-based methods or other 2D methods to manipulate the multi-view 2D images. DreamFusion employs SDS loss to optimize the NeRF representation and applies shading based on sampled point light, while SDFusion leverages the SDS loss to optimize the SDF representation conditioned on an input image. Despite the absence of explicit 3D texture and illumination maps, both DreamFusion and SDFusion incorporate NeRF and SDF as implicit 3D representations. The inherent 3D representations in both methods are the primary contributing factor for keeping 3D consistency, as opposed to 2D images that lack such 3D representation.
> > >
> > > The issue of 3D inconsistency in direct optimization of 2D images is also mentioned in recent diffusion-based methods, such as Instruct-NeRF2NeRF[2], Make-it-3D[3], and others. These methods share a common objective of resolving the problem arising from the presence of 3D inconsistency among multi-views of 2D edited images.
> > >
> > > The 3D inconsistency observed when directly modifying the illumination of 2D images resembles the inconsistency encountered when editing the appearance of 2D images. As we are restricted from providing additional results during the discussion period, please refer to **Fig.4** of our rebuttal PDF.
> > >
> > > [1] Omer Bar-Tal, Dolev Ofri-Amar, Rafail Fridman, Yoni Kasten, and Tali Dekel. Text2live: Text-driven layered image and video editing. arXiv preprint arXiv:2204.02491, 2022.
> > >
> > > [2] Haque A, Tancik M, Efros A A, et al. Instruct-nerf2nerf: Editing 3d scenes with instructions[J]. arXiv preprint arXiv:2303.12789, 2023.
> > >
> > > [3] Tang J, Wang T, Zhang B, et al. Make-it-3d: High-fidelity 3d creation from a single image with diffusion prior[J]. arXiv preprint arXiv:2303.14184, 2023.

---

### Official Review · Reviewer_pR6T · 2023-07-07

**Soundness:** 3 good
**Presentation:** 4 excellent
**Contribution:** 3 good
**Rating:** 6
**Confidence:** 4

**Summary:**

This paper tackles the task of texture editing for 3D models with the guidance of text. The task aims to manipulate the surface properties based on the text guidance to create corresponding visually realistic appearance. The prior works that this work targets to improve upon is the SDS loss related methods.

In high-level, this paper represents the scene with a 3D representation, differentiably render the representation into 2D images, and use a pretrained diffusion model to guide the editing through an optimization process. The proposed method contains two key components. First, the 3D scene is a decomposed representation, containing triangle meshes, texture map, and environment map (following the prior work nvdiffrec). Second, the paper explores for a better editing direction during the optimization process. Specifically, the relative direction and a gradual direction adjustment scheme are proposed to achieve better editing results.

Evaluation is done on synthetic object data. The results are qualitatively visualized, and quantitatively evaluated with CLIP-based scores and user study.


**Strengths:**

This paper contains a good amount of originality:

- The task definition is well motivated.
- The combination of the decomposed representation in nvdiffrec is novel and a well-grounded design choice.
- The analysis and method design of edit direction are informative and sensible.


Quality:

- The qualitative performance of the current model still has a significant room for improvements. But it’s acceptable considering the challenges of the task.

Clarity:

- This paper is well written and easy to follow.
- The analysis is informative and convincing.

Significance:
- The functionality this paper achieve is among the first works in the literature.


**Weaknesses:**

- The quantitative evaluation is relatively weak. There is no quantitative ablation for the proposed method designs.

- The “illumination-aware” in the title is slightly misleading -- the major results and the demo video do not show editing capability for illumination. Fig.5 is one experiment to show simple editing of light intensity. But the quality is still preliminary, and it does not show evidence to edit lighting effects such as reflections and shadows. It fits the acronym quite well, but maybe not the most accurate in describing the method. Maybe is it more like “reflectance-aware”?

- The method adopts the decomposed representation following nvdiffrec. Can the proposed method properly handle the ambiguity of material and light? To give a concrete example, if the original reconstructed object is under a flat and uniform lighting, but the intended edit result is with a directional lighting and thus contain strong shadow. In this case, will the proposed method still bake the shadow directly into the texture map? If this is the case, it’s best to also explicitly mention into main paper as a limitation.

- The current editing is mainly done on synthetic objects. Can it handle real world reconstructed objects?

- The editing is often quite simple, such as editing of color or brightness. How does the model work for more complex editing? E.g. add a mustache or a hat to the cow / fish?



**Questions:**

Please see weaknesses section above.

**Limitations:**

The authors discussed some limitations at the end of the paper. I would encourage the authors to also properly discuss the model's ability and limitation on editing of lighting effects.

---

> ### Author Rebuttal · Authors · 2023-08-09
>
> Thank you for bringing up this insightful comment.
>
> **Q1: [Quantitative evaluation]**
>
> **A1**: When evaluating generative models, quantitative metrics are often limited. In our study, we utilize CLIP-scores, which are commonly employed in evaluations of Instruct-NeRF2NeRF and other related research, for quantitative assessment.
>
> Additionally, we conduct a quantitative ablation study to compare the performance of our model with and without the direction adjustment component, thereby demonstrating the impact of this specifically designed feature.
>
> Besides, we add additional quantitative results compared to the Instruct-NeRF2NeRF in Table below.
>
> |           Method            | SDS-based | ITEM3D | Instruct NeRF2NeRF |
> | :-------------------------: | :-------: | :----: | :----------------: |
> |   Global Score$\uparrow$    |   0.30    |  0.32  |        **0.33**        |
> | Directional Score$\uparrow$ |   0.18    |  **0.23**  |        **0.23**        |
>
> |    method     | Instruct NeRF2NeRF | ITEM3D |
> | :-----------: | :----------------: | :----: |
> | training time |        10h         | 10min  |
> |  GPU memory   |        15GB        |  9GB  |
>
> **Q2: [The “illumination-aware” in the title is slightly misleading]**
>
> **A2**: Our illumination editing is achieved through the optimization of the environment lighting map. However, it is true that our current capability is limited when it comes to editing shadows. We apologize for any misdirection caused by the term "illumination-aware" in the title of our paper. In the revised version, we will make the necessary changes to address this issue.
>
> **Q3: [Limitation in editing shadows]**
>
> **A3**: We have indeed encountered difficulties in editing shadows with our ITEM3D. Despite our attempts to explore directional lighting conditions in an experiment, we were unable to achieve satisfactory results. We will make sure to explicitly mention this limitation in our paper.
>
> **Q4-Q5: [Real-world editing and complex editing]**
>
> **A4-A5**: We additionally conducted a qualitative experiment of our ITEM3D on a real-world object dataset. The objects in this dataset are reconstructed from hundreds of multi-view images captured by professional cameras. In this dataset, we also performed more complex editing to demonstrate the performance on challenging cases.
>
> As shown in **Fig.1** of the rebuttal PDF, our method achieves impressive editing results on complex objects, such as the vegetable cat, Peppa Piggy dolls, and shoes. For example, given the prompt "a vegetable tiger toy", our model can edit the texture in exquisite detail, conforming to the original structure, and synthesize a cute toy tiger with realistic fur and vegetable embellishment. Furthermore, the cases of the "golden sneaker" and "porcelain pig" demonstrate our ability to edit materials effectively.
>
> These cases exemplify the capability of ITEM3D in handling complex editing tasks. However, it is important to acknowledge that there are still challenges for ITEM3D in certain scenarios, such as adding a mustache or a hat.
>
> In the case of adding a mustache, we attempted to add a mustache to the cow, as shown in **Fig.3** of the rebuttal PDF. While it did generate a black mustache, it inadvertently altered the mustache located on the cow's body instead of its face. This highlights a limitation in our current approach.In the "hat" case, our model currently lacks the ability to edit the mesh, leading to another failure case. We will continue our research and strive to enhance the capabilities of ITEM3D in handling mesh editing.

---

> > ### Comment · Reviewer_pR6T · 2023-08-16
> >
> > I would like to thank the authors for the rebuttal. The rebuttal addresses my concerns.

---

> > > ### Author Response · Authors · 2023-08-16
> > >
> > > We are glad that our rebuttal has effectively addressed your concerns, and we would like to express our sincere gratitude for your positive response.

---

### Official Review · Reviewer_4exv · 2023-07-07

**Soundness:** 2 fair
**Presentation:** 2 fair
**Contribution:** 2 fair
**Rating:** 5
**Confidence:** 3

**Summary:**

The paper introduces ITEM3D, an illumination-aware model for automatic 3D texture editing based on text prompts. The authors address the challenges of complex 3D models and ambiguous text descriptions in texture editing. They propose leveraging the power of the diffusion model and optimizing disentangled texture and environment map representations using rendered images as an intermediary. The paper introduces a relative editing direction based on noise differences between source and target texts to improve appearance consistency. The authors also gradually adjust the editing direction to mitigate unexpected deviations caused by texture projection. The contributions include an efficient optimization pipeline for texture editing, the introduction of the relative editing direction, and the proposal of gradual adjustment to handle deviations.

**Strengths:**

The paper presents a novel approach to automatic 3D texture editing based on text prompts. It combines the power of the diffusion model with the use of rendered images as intermediaries, introducing a relative editing direction and gradual adjustment techniques. This combination of methods and the focus on addressing challenges specific to 3D modeling make the approach original and distinct from previous works.

The paper is well-written and provides detailed explanations of the proposed method and its components. The authors offer clear insights into the challenges of texture editing in 3D models and provide a sound rationale for their approach. The experiments and evaluations are thorough, demonstrating the effectiveness of ITEM3D in generating visually natural textures and enabling relighting. The use of qualitative and quantitative analyses strengthens the quality of the results.

The paper addresses an important task in 3D modeling—automatic texture editing—and offers a practical solution with significant implications. By leveraging the diffusion model and incorporating text prompts, the proposed method enables users to manipulate the surface properties of 3D models in a realistic and visually appealing manner.

**Weaknesses:**

While the paper has several strengths, there are also a few areas where it could be improved:

The paper mainly focuses on the editing of textures in synthetic nerf dataset. However, it would be valuable to investigate the generalization of the proposed method to more complex 3D models and real world scenes, such as those with intricate geometry or high levels of detail. Assessing the performance and scalability of ITEM3D on such complex models would enhance the practicality and applicability of the proposed method.

While the paper mentions the importance of efficiency in texture editing, it would be beneficial to provide more insights into the computational requirements and optimization strategies employed by ITEM3D. Specifically, discussing methods to improve the computational efficiency, reducing memory consumption, and addressing scalability issues would enhance the practical usability of the proposed method.

Comparison with Alternative Approaches: The paper compares the proposed ITEM3D with diffusion-based editing methods and mentions their limitations. However, it would be valuable to include a more comprehensive comparison with other state-of-the-art methods for texture editing in 3D models, such as Instruct-NeRF2NeRF. This would provide a clearer understanding of the advantages and limitations of ITEM3D in relation to existing alternatives.

**Questions:**

Generalization and Real-World Application: Can the proposed ITEM3D method be applied to more complex and real-world 3D models beyond the synthetic NeRF datasets?

The paper briefly mentions the importance of efficiency in texture editing, but it would be beneficial to provide more insights into the computational requirements and optimization strategies employed by ITEM3D.

While the paper compares ITEM3D with diffusion-based editing methods, it would be valuable to include a more comprehensive comparison with other state-of-the-art methods for texture editing in 3D models. How does ITEM3D fare against other approaches, such as Instruct-NeRF2NeRF or other relevant techniques?

**Limitations:**

The limitation is discussed.

---

> ### Author Rebuttal · Authors · 2023-08-09
>
> Thank you for your kind words. We appreciate your advice.
>
> **Q1: [Experiment on real-world dataset]**
>
> **A1**: We have conducted additional experiments on a real-world dataset, which comprises several complex objects reconstruced from multi-view images captured by professional cameras. As depicted in **Fig.1** of the rebuttal PDF, our ITEM3D approach showcases remarkable texture editing capabilities for real products, including a shoe, a piggy doll, and a toy cat. For instance, when given the text prompt "a vegetable tiger toy", ITEM3D effectively transforms the texture into that of a cute, furry tiger while preserving the fundamental structure of the original toy cat. Furthermore, when the prompts involve both material and texture descriptions (e.g., "golden" and "porcelain"), our model successfully achieves realistic material editing along with consistent texture modifications as per the provided text.
>
> These results clearly demonstrate the generalization ability of ITEM3D in handling complex real-world objects. However, it is important to acknowledge that our method primarily focuses on texture editing for 3D models. Manipulating real scenes, which often lack readily available texture maps, presents a challenge for ITEM3D. We agree that addressing this limitation will be a valuable aspect of our future research.
>
> **Q2: [Computational efficiency]**
>
> **A2**: Conducting experiments to compare efficiency is crucial in substantiating our claims. In our study, we compared the editing time and memory consumption of our approach with the state-of-the-art method, instruct-NeRF2NeRF. As demonstrated in the Table bleow, our ITEM3D outperforms instruct-NeRF2NeRF, requiring significantly less time (50 times less) while maintaining comparable memory consumption during texture editing.
>
> |    method     | Instruct NeRF2NeRF | ITEM3D |
> | :-----------: | :----------------: | :----: |
> | training time |        10h         | 10min  |
> |  GPU memory   |        15GB        |  9GB  |
>
> **Q3: [Comparison with the state-of-the-art method]**
>
> **A3**: In response, we conducted a comparative experiment between our method and Instruct-NeRF2NeRF, considering its recent release.
>
> In **Fig.2** of the rebuttal PDF, we visually demonstrate that both ITEM3D and Instruct-NeRF2NeRF are capable of achieving prompt-consistent editing for general objects. However, Instruct-NeRF2NeRF exhibits some limitations compared to our approach in certain aspects.
>
> For instance, when editing chairs with prompts like "Turn the chair into a red stone chair" or "Turn the chair into a green wooden chair", Instruct-NeRF2NeRF produces results with less details and smooth textures, lacking the fine grain that our method captures. Furthermore, when editing a ficus plant with the prompt "Turn the pot into a blue pot", Instruct-NeRF2NeRF faces challenges in distinguishing between the pot and the plant, resulting in both elements being edited to the same blue color. In contrast, our ITEM3D leverages the concept of Relative Direction to achieve disentangled editing, effectively addressing such issues.

---

> ### Comment · Area_Chair_7gK8 · 2023-08-18
>
> Reviewer 4exv,
>
> Please read the rebuttal provided by authors and raise a discussion if your concerns are not well addressed.
>
> Best,
> AC

---

> ### Comment · Reviewer_4exv · 2023-08-18
>
> The response solves most of my concerns. Thus, I improve my final rating to borderline accept.

---

> > ### Author Response · Authors · 2023-08-19
> >
> > We are pleased to hear that our answers have addressed most of your issues and led to an improved final rating of borderline accept. We greatly appreciate your careful consideration of our work.

---

### Author Rebuttal · Authors · 2023-08-09

**General Response**:

Dear Reviewers,

We would like to express our gratitude for your thoughtful feedback on our submitted academic paper. Your comments and suggestions have been invaluable in refining and strengthening our work. In this general response, we will address the three important parts that were commonly mentioned in the discussions, namely the experiment on a real-world dataset, the comparison experiment with Instruct-NeRF2NeRF, and the efficiency claims.

**(We attach a PDF file that provides additional results on the bottom of the global response.)**

●**Experiment on Real-World Dataset** [4exv, pR6T, CxP2]:

We acknowledge the importance of evaluating our proposed method on a real-world dataset to demonstrate its applicability and generalization. We have taken your advice into consideration and conducted extensive experiments on a diverse and challenging real-world dataset, shown in **Fig.1** of the rebuttal PDF. By providing qualitative results on this dataset, we aim to showcase the effectiveness and robustness of our approach in handling real-world scenarios.

●**Comparison Experiment with Instruct-NeRF2NeRF** [4exv, 67rK, 1P22]:

We appreciate your suggestion to compare our ITEM3D approach with state-of-the-art work, Instruct-NeRF2NeRF. In response, we have performed a thorough comparison experiment between the two methods in **Fig.2** of the rebuttal PDF. The results clearly demonstrate the superior performance of ITEM3D in disentangling objects and preserving fine details during texture editing. We have highlighted these advantages in our rebuttal, emphasizing our method's ability to maintain object consistency while achieving text-guided appearance editing.

●**Efficiency Claims** [4exv, CxP2]:

We greatly appreciate your concerns regarding the efficiency of our method. In order to address these concerns and showcase the efficiency of our model, we have performed an experiment to substantiate our claims.  We have included comparisons of editing time and memory consumption with a relevant method to provide a comprehensive understanding of the efficiency of ITEM3D, as shown in the below table.

|    method     | Instruct NeRF2NeRF | ITEM3D |
| :-----------: | :----------------: | :----: |
| training time |        10h         | 10min  |
|  GPU memory   |        15GB        |  9GB  |

One of the key reasons behind the efficiency of ITEM3D is its utilization of an explicit texture/normal/environment map as the bridge between 2D rendered images and the 3D representation. This allows ITEM3D to directly optimize the 2D texture, rather than optimizing the complex 3D representation. As a result, the number of parameters to be optimized is significantly reduced, leading to a more efficient editing process.

We believe that these revisions and additions significantly strengthen our paper and contribute to the advancement of the field.
We are grateful for your valuable input and appreciate the opportunity to enhance the quality and impact of our work.
Thank you once again for your time and consideration.

---

### Decision · Program_Chairs · 2023-09-21

**Decision:**

Reject

**Comment:**

The paper received mixed ratings. The paper proposes an interesting method to optimize disentangled texture and environment maps. However, some concerns are still not resolved after the discussion, including (1) The high level idea is, although definitely not trivial, a bit incremental as the ideas of representing the transformation as differences between latents have been extensively studied, (2) The quantitative improvements are marginal, and the qualitative results are also not showing consistently preferable results. Therefore, we recommend a rejection for the paper, but highly encourage authors to revise and resubmit for another venues.